# Filter Design for Laser Inertial Navigation System Based on Improved Pigeon-Inspired Optimization

**Zhihua Li** [1,2,*], **Lin Zhang** [2] **and Kunlun Wu** [2]

1 School of Automation Science and Electrical Engineering, Beihang University, Beijing 100083, China
2 Beijing Aerospace Times Laser Inertial Technology Company, Ltd., Beijing 100094, China
* Correspondence: by1703133@buaa.edu.cn

**Abstract:** The laser gyroscope of Laser Inertial Navigation System (LINS) eliminates the influence of the locked zone with mechanical dither. The output information of laser gyroscopes must be filtered before use to eliminate vibration noise. Laser gyroscope filters are designed according to the instrument accuracy, calculation capacity, vibration frequency, system dynamic characteristics, and other indicators. In this paper, a pigeon-inspired optimization (PIO) method is proposed for use in filter design. The PIO method can flexibly design filters with excellent performance according to the indicator requirements. In the method, the constraints and indicators of the amplitude, phase and order of the LINS filter are firstly confirmed according to the application requirements; then, the objective function is established, and the parameters to be optimized of the PIO are set according to the order of the filter; finally, the PIO method is used to obtain filter parameters that can satisfy the constraints and achieve better performance. Referring to the idea of biological evolution mechanisms, we propose a new improved pigeon-inspired optimization method based on natural selection and Gaussian mutation (SMPIO), which can obtain more stable results and higher accuracy. In the SMPIO method, the particle swarm is firstly selected by natural selection, that is, the particles are sorted according to the fitness function, and some particles with poor fitness are replaced by those with better fitness; then, all particles are subjected to Gaussian mutation to obtain a better global optimum. SMPIO method can flexibly design filters according to the comprehensive requirements of laser gyro performance and navigation control indicators, which cannot be achieved by traditional filter design methods; the improvement based on natural selection and Gaussian mutation enables SMPIO to have faster convergence speed, and higher accuracy.

**Keywords:** inertial navigation; laser gyroscope; filter; improved pigeon-inspired optimization (IPIO)

## 1. Introduction

For the convenience of introduction, some abbreviations are used according to relevant standards. The main abbreviations of this article are shown in Table 1.

An inertial navigation system (INS) is a self-contained device consisting of an inertial measurement unit (IMU) and computational unit. The IMU is typically made up of 3 accelerometers and 3 gyroscopes and measures the system's angular rate and acceleration [1,2]. The computational unit used to determine the attitude, position, and velocity of the system based on the raw measurements from the IMU given an initial starting position and attitude. LINS are composed of three laser gyroscopes and three accelerometers.

Laser gyroscopes have the advantages of high precision, a large dynamic range, good reliability, and fast start-ups. The laser gyroscopes used in laser inertial navigation system are a navigation-level gyroscopes and are widely used in various fields. Yu [3] discussed the mechanical dither device of the ring laser gyroscope and its improvement. Banerjee [4] discussed and researched the dither removal techniques of laser gyroscopes; Chuang [5] adopted Wavelet packet analysis to filter the output signal of the ring laser gyroscope; Wu [6] summarized the application of strapdown inertial navigation technology

in the measurement; Additionally, Kuznetsov [7] introduced the autonomous and precise navigation of laser gyroscope inertial navigation. Mechanically vibrating laser gyroscopes were the first laser gyroscopes to be used in practice, and they are also the most widely used laser gyroscopes at present. They use alternating mechanical vibrations that cause the gyroscope to be outside of the locked zone, thus reducing lock-in errors [8,9]. The digital filtering method is used by laser gyroscopes to demodulate their output signals.

**Table 1.** The main abbreviations of this article.

| Acronyms | Full Forms |
|---|---|
| INS | inertial navigation system |
| IMU | inertial measurement unit |
| LINS | laser inertial navigation system |
| IIR filter | infinite impulse response filter |
| FIR filter | finite impulse response filter |
| EA | evolutionary algorithm |
| PIO | pigeon-inspired optimization |
| IPIO | improved pigeon-inspired optimization |
| SMPIO | pigeon-inspired optimization with natural selection and Gaussian mutation |
| GA | genetic algorithm |
| PSO | particle swarm optimization |

Scholars have studied the filter method for laser gyroscopes, for example, Mark [10] developed a high-speed moving average filter (1 k Hz) to reduce the effect of quantization and dither on gyro test data so as to enable the random walk coefficient of the instrument to be determined to a high precision in a short space of time; Regimanu [11] developed two types of multistage digital filters, namely, BBB and BCO filters to reduce the dither signal to an acceptable level; Yan [12] researched a scheme using an LMS adaptive filtering algorithm to meet the speed and precision requirements of laser gyro demodulation aerospace fields; Chen [13] adopted a combined digital filter consisting of an IIR filter and an FIR filter to remove the dither signal; Fan [14] introduced a novel dither-controlling method without external feedback; and Regimanu [15] used a modified Stockwell transform (MST) to filter the dither signal. However, FIR and IIR filters are still widely used in engineering.

According to the network structure or impulse response per unit, digital filters can be divided into infinite impulse response (IIR) filters and finite impulse response (FIR) filters. Traditional filter design methods mainly include the window function method, the frequency sampling method, and the best uniform approximation method. An intelligent optimization algorithm has been proposed for the design of filters, and good results have been obtained. The intelligent algorithm used for filter design includes particle swarm optimization, cat swarm optimization, and artificial bee swarm optimization.

In terms of the PSO algorithm, Dash [16] proposed population-based derivative free diffusion particle swarm optimization (DPSO) algorithms to estimate the parameters of IIR system; Eswari [17] used an improved particle swarm optimization (IPSO) to identify an infinite impulse response (IIR) system based on the concept of error minimization; Liu [18] designed an IIR filter using an improved adaptive inertia weight particle swarm optimization (PSO) algorithm to reduce the computing costs and improve the convergence speed of the filter weight; Sarangi and Panda [19,20] proposed a crossover cat swarm optimization algorithm for the identification of unknown IIR system; Karaboga [21] proposed a new method based on the artificial bee colony (ABC) algorithm for the design of digital IIR filters; Upadhyay [22] applied a population-based evolutionary algorithm methodology called the opposition-based harmony search (OHS) algorithm for the optimization of the system coefficients of adaptive infinite impulse response system identification problems; Dash and Upadhyay [23,24] proposed an improved firefly algorithm for IIR system optimization; and Yadav [25] designed an FIR filter using whale optimization.

The previous article first introduced the laser inertial navigation system and its application requirements for filter design. Then, the current research status of laser gyroscope filter design and the demand for higher-performance filters are introduced. Then, some research progresses on EA methods and their applications in filter design are presented. Compared with the more complex filtering methods mentioned in some literature, engineering applications still tend to be simple and practical filters with a small amount of calculation, and it is necessary to flexibly perform amplitude and phase synthesis indicators according to control requirements. Compared with the EA method mentioned in some literature for filter design, a technique with faster convergence speed and better stability can be found. In this study, we will use a new, improved PIO method to design filters to achieve better performance according to application requirements.

## 2. Design Method of LINS Filter Based on SMPIO

### 2.1. LINS Filter

The mechanically vibrating laser gyroscope is shown in Figure 1. The laser gyro shakes the yellow glass block back and forth around the central axis through the dither motor in the center. The dither motor consists of 8 spokes. The power to vibrate is provided by piezoelectric ceramics bonded to the spokes. Piezoelectric ceramics are driven by voltage signals to control the dither of the laser gyroscope. The shaking mechanism of the laser gyroscope is an essential part of its function and accuracy. Scholars have conducted a lot of research on it, such as Barantsev [26] who analyzed the impact of mechanical vibration of ring laser gyroscopes on the accuracy of attitude determination in a strapdown INS. Lee [27] studied the elimination of the lock-in effect of laser gyroscopes by phase wrapping/unwrapping. Aviev [28] developed a photoelectric system to measure the dither parameters of the laser gyroscope. The gyroscope operates outside the locked zone most of the time when the positive and negative alternating vibrations are added to the mechanically vibrating laser gyroscope. Since the duration of the mechanically vibrating laser gyroscope inside the locked zone is very short, the error caused by the locked zone is greatly reduced, even if the input angular rate is small. For mechanically vibrating laser gyroscopes, its output information includes not only the angular rate information of the external input, but also the angular rate information of the vibration signal (as shown in Figure 2). Therefore, the output information must be demodulated to eliminate the vibration angular rate information. When sinusoidal alternating vibrations are used, the total output angular rate of the laser gyroscope containing the external input angular rate information is as follows:

$$\Omega_p = \Omega_r + \Omega_D \sin(2\pi f_D t) \tag{1}$$

where $\Omega_D$ is the maximum angular rate of dither; $f_D$ is the dither frequency; and $\Omega_r$ is the angular rate to be measured.

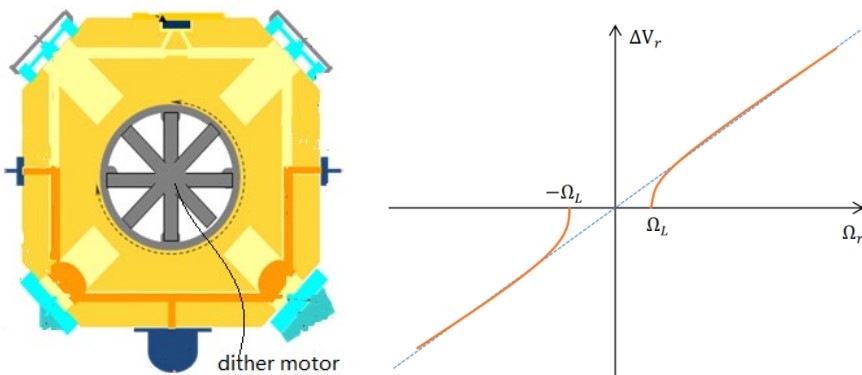

**Figure 1.** Laser gyro and its lock-in effect.

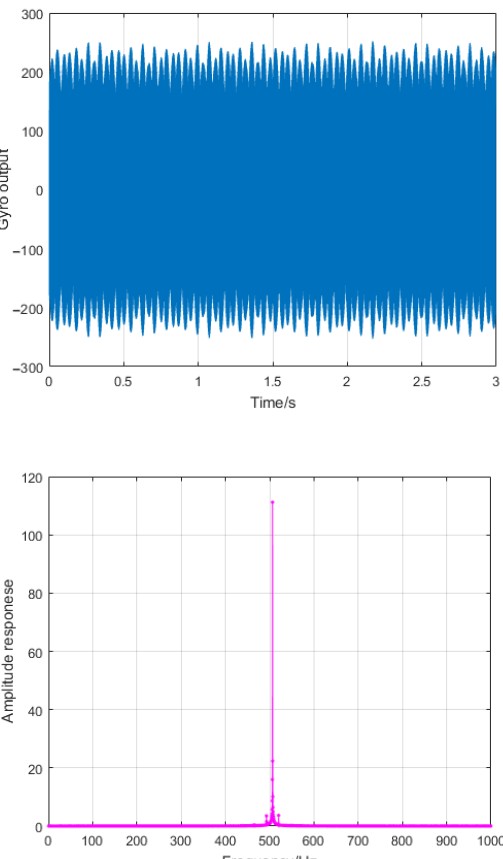

**Figure 2.** The original information and spectrum analysis results of the laser gyro ($f_s = 2$k Hz, $f_D = 515$ Hz).

The output signal of the laser gyroscope includes the frequency difference $\Delta V_r$ caused by the angular rate $\Omega_r$ to be measured and the frequency difference $\Delta V_p$ caused by the vibration angular rate $\Omega_D$. $\Delta V_p$ needs to be eliminated by the filtering and demodulation method. The effective angular rate input spectrum range of the inertial navigation is usually 0~100 Hz, while the vibration frequency of mechanically vibrating laser gyroscopes is generally above 300 Hz [29]. According to the sampling theorem, the output signal of laser gyroscopes can be collected at a sampling frequency that is more than 2 times higher than the vibration frequency, and filtered by low-pass digital filtering. In this way, the vibration noise introduced by the vibration signal and the high-frequency noise caused by other factors can be eliminated. According to the above situation, the low-pass digital filtering method is often used to demodulate the output signal of a gyroscope.

The laser inertial navigation system must perform error compensation calculations of inertial instruments (gyroscopes and accelerometers), filter calculations, dynamic error calculations, fault diagnosis calculations, and inertial navigation solutions in actual application. When the inertial navigation system is combined with the satellite navigation system, integrated navigation calculations are required. When the inertial navigation system is combined with the aircraft's vision system, visual navigation calculations are required. The computer of the inertial navigation system may need to perform the above calculations simultaneously, and the calculation speed is necessary to be very fast (the update frequency of some calculations is usually 2000 Hz or higher), so specific requirements are put forward for the calculation capacity. IIR filters have high calculation accuracy and can satisfy the calculation requirements with a lower order, which has a smaller number of filter coefficients. IIR filters are available for laser inertial navigation system with limited calculation capacity.

The form of the *m*-order low-pass IIR filter used by LINS is as follows:

$$y(n) = -\sum_{k=0}^{m} a_k y(n-k) + \sum_{k=0}^{m} b_k x(n-k) \qquad (2)$$

where $x(n)$ is the input sequence; $y(n)$ is the output sequence; $a_k$, $b_k$ is the filter coefficient of the digital filter; $m$ is the filter order, and filters with a smaller $m$ are called lower order filters.

Generally, the low-pass digital filter in the ideal state is used as the reference filter, and the designed filter is compared with the reference filter to evaluate the performance of the filter. The performance evaluation indicators of the digital filter are often expressed by the allowable error difference of the frequency response amplitude. Taking the low-pass digital filter as an example, the frequency response can be divided into three parts: a pass band, a transition band, and a stop band. The ideal low-pass filter amplitude curve should satisfy the following conditions: the amplitude in the pass band should be constant; the amplitude in the stop band should be 0; and the width of the transition band should be 0, that is:

$$\begin{cases} \left| H_p\left(e^{jw}\right) \right| = 1, & \omega \leq \omega_p \\ \left| H_p\left(e^{jw}\right) \right| = 0, & \omega_s \leq \omega \leq \pi \end{cases} \tag{3}$$

where $\omega_s$ is the cutoff frequency of the stop band; $\omega_p$ is the cutoff frequency of the pass band; and the bandwidth of the transition band is $(\omega_s - \omega_p)$.

In fact, the above three conditions cannot be absolutely satisfied, and there will be certain errors, that is:

$$\begin{cases} 1 - \delta_p \leq \left| H_d\left(e^{jw}\right) \right| \leq 1 + \delta_p, & \omega \leq \omega_p \\ \left| H_d\left(e^{jw}\right) \right| \leq \delta_s, & \omega_s \leq \omega \leq \pi \end{cases} \tag{4}$$

where $\delta_p$ is the allowable error in the pass band; the amplitude is infinitely close to 0 in the stop band; and the allowable error is $\delta_s$.

### 2.2. SMPIO Method

The pigeon-inspired optimization is a new swarm intelligence optimizer inspired by the hidden mechanism behind the remarkable navigation capacity of homing pigeons [30]. Zhong [31] pointed out that the PIO algorithm has better global optimization ability. Nath [32] applied the PIO algorithm and proved it could maintain high stability and accuracy. Some studies demonstrate that IPIO can further improve performance. He [33] introduced quantum evolution into the PIO algorithm to solve the problem of falling into the local optimum. Duan [34] incorporated the mutant mechanism to strengthen the exploration capability of PIO. PIO and its variants have been widely used in various fields, from combinatorial optimization to multi-objective optimization. Cui [35] adopted an improved PIO algorithm (ImMAPIO) to solve the multi-objective optimization problem. Li [36] used PIO to identify INS sensor errors from navigation data. Peng [37] used PIO for unmanned aerial vehicle (UAV) swarm cooperative control. The map and compass operator and the landmark operator are the two different operators of the PIO algorithm. Compared with EA algorithms such as GA and PSO, PIO shows higher optimization accuracy and faster convergence speed [38,39].

The PIO algorithm model is as follows:

$$\begin{gathered} X_i = [x_{i1}, x_{i2}, \ldots x_{iD}] \\ V_i = [v_{i1}, v_{i2}, \ldots v_{iD}] \\ V_i^{N_c} = V_i^{N_c-1} e^{-R \cdot N_c} + rand\left( X_{gbest} - X_i^{N_c-1} \right) \end{gathered} \tag{5}$$

$$X_i^{N_c} = X_i^{N_c-1} + V_i^{N_c} \tag{6}$$

where $V_i = [v_{i1}, v_{i2}, \ldots v_{iD}]$ is the velocity, and it is updated according to Equation (5); $X_i = [x_{i1}, x_{i2}, \ldots x_{iD}]$ is the position, and it is updated according to Equation (6); $R \in (0 \sim 1)$ is the map and compass factor; $rand \in (0 \sim 1)$ is a random number; $X_{gbest}$ is the global optimal position obtained by comparing the positions of all the pigeons after $N_{c-1}$ iteration cycles; $N_c$ is the current number of iterations [36].

The mechanism of biological evolution includes reproduction, mutation, genetic recombination, natural selection, and other evolutionary processes. Referring to the idea of biological evolution mechanisms, we propose an improved pigeon-inspired optimization method based on natural selection and Gaussian mutations to further improve the optimization effect of the algorithm. The specific improvement methods are as follows:

1.      Natural selection of the particle swarm:

For the minimum value optimization problem, the $n_{total}$ particles of the current generation are sorted in the order of fitness, from small to large, and the $n_{select}(n_{select} < n_{total})$ particles with the larger fitness are eliminated and replaced by the position and speed of the $n_{select}$ particles with the smaller fitness, that is:

$$
\begin{aligned}
X^{N_c}_{i(sortx(exIndex \sim n_{total}))} &= X^{N_c}_{i(sortx(1 \sim n_{select}))} \\
V^{N_c}_{i(sortx(exIndex \sim n_{total}))} &= V^{N_c}_{i(sortx(1 \sim n_{select}))}
\end{aligned}
\tag{7}
$$

where $sortx$ is the sequence number of $n_{total}$ particles sorted from small to large according to fitness; $sortx(1 \sim n_{select})$ is the sequence number of current-generation $n_{select}$ particles; and $sortx(exIndex \sim n_{total})$ is the sequence number of next-generation $n_{select}$ particle.

2.      Gaussian mutation of the particle swarm:

After the particles are naturally selected and the speed and position are updated according to the PIO method, Gaussian mutation should be applied to the particles to prevent them from falling into local optimization or to provide them with better optimization performance, that is:

$$
X^{N_c}_i = X^{N_c}_i + randn(1, dim) * k
\tag{8}
$$

where $dim$ is the dimension of the particle; $randn(1, dim)$ is a random number of Gaussian distribution and has the same dimension as $X^{N_c}_i$; and $k$ is the coefficient, which is confirmed according to the dispersion of the particle's position when the PIO method is applied.

### 2.3. Filter Design Method Based on SMPIO

Based on the above analysis, we propose the improved PIO method with natural selection and Gaussian mutation for the filter design. Considering the general dynamic characteristic requirements of the medium and low-speed operating environments of drones, the fitness function mainly takes the amplitude error as the reference standard and the phase parameter as the verification indicator. Taking the *m*-order low-pass IIR filter as an example, the dimension of the particle number is *2m*−1, and, with $a_0 = 1$, $\frac{\sum_{k=0}^{m} b_k}{\sum_{k=0}^{m} b_k} = 1$ is satisfied; the fitness function is:

$$
fitness = \left\{ \sum_{i=1}^{L} \left[ H_d\left(e^{j\omega i}\right) - H_P\left(e^{j\omega i}\right) \right]^2 \right\}^{1/2} * \max\left( \left| H_d\left(e^{j\omega i}\right) - H_P\left(e^{j\omega i}\right) \right| \right)
\tag{9}
$$

where *i* = *1~L* is the selected sampling point; $H_d(e^{jwi})$ is the amplitude of the designed filter at sampling point *i*; and $H_P(e^{jwi})$ is the amplitude of the ideal filter at sampling point *i*.

The steps of Filter Design Method Based on SMPIO shown in Figure 3 are as follows:

1.      A set of initial values of particle swarm is randomly generated by Monte Carlo method. The fitness function value of the particle swarm is calculated and compared to find the globally optimal.
2.      The fitness function of the IIR filter, that is Formula (9), is calculated as follows:

- Obtain the coefficient $a_k$, $b_k$ of the IIR digital filter according to the particle swarm position data.
- Calculate the amplitude response $H_d(e^{j\omega i})$ of each sampling point of the designed filter according to $a_k$, $b_k$, then compare it with the amplitude response $H_P(e^{j\omega i})$ of the ideal filter, and then calculate the fitness function according to Formula (9).

3.  Natural selection of the particle swarm: Sort the current generation of particles according to their fitness, from small to large, and update the speed and position of $n_{select}$ particles with the largest fitness according to Formula (7).
4.  Update the particle swarm velocity according to Formula (5) and update the particle swarm position according to Formula (6).
5.  Gaussian mutation of the particle swarm: Mutate the position of the particle swarm according to Formula (8);
6.  Calculate the fitness function value of the particle swarm after position update, compare and record the global optimal particle position and fitness value.
7.  Determine whether the specified maximum number of iteration steps is reached. If not, go to step 3, and continue iterative calculation; if so, end the process and save the calculation results.

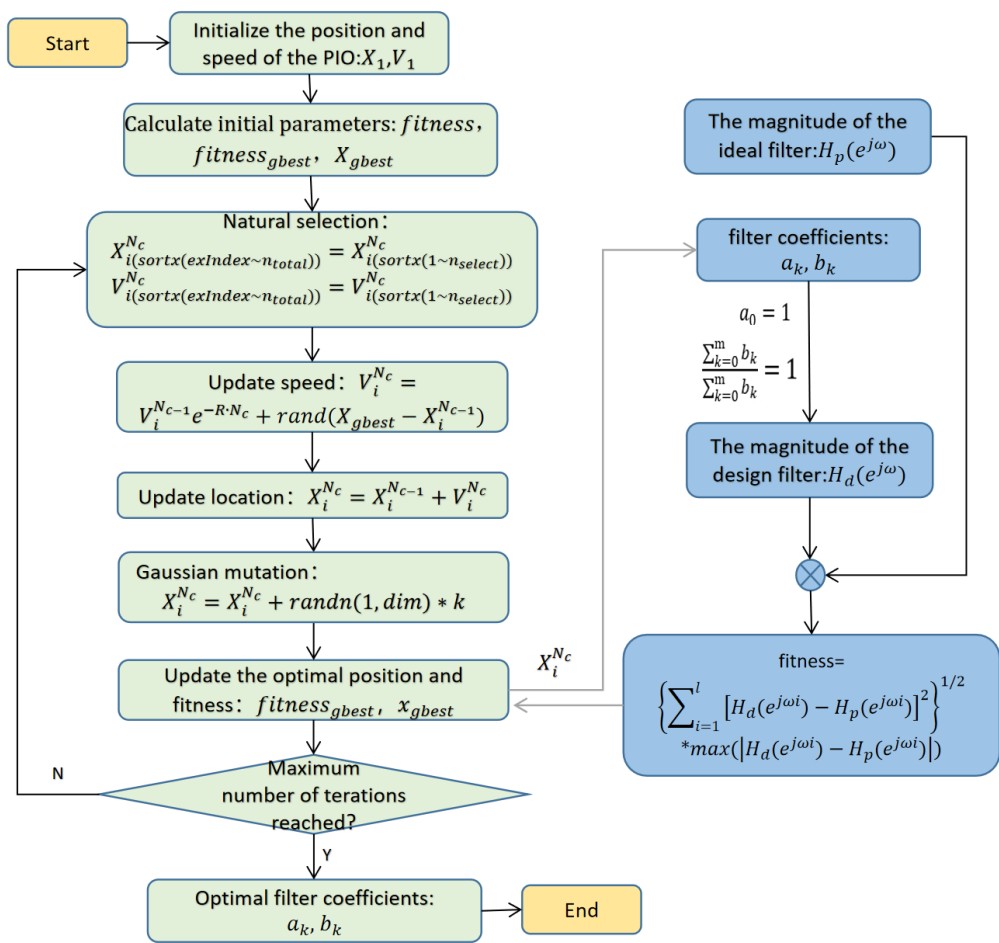

**Figure 3.** Flow chart of filter design method based on SMPIO.

In addition, for some aircraft with large maneuvering, the INS filter not only requires the amplitude to meet certain indicators, but also requires a small phase to meet the needs of real-time control of the aircraft. LINS has high requirements for accuracy and real-time performance when it is used for flight attitude control. The filter design problem with comprehensive amplitude and phase requirements is a multi-objective optimization problem. The Chebyshev 1 filter, the Chebyshev 2 filter, and the elliptical filter use the amplitude as the design indicators and do not consider the design indicators of the phase. The SMPIO is suitable for the filter design considering the comprehensive indicators of amplitude and phase. The process is shown in Figure 3 when the SMPIO method is used to

comprehensively consider the amplitude and phase indicators to design the filter, but the fitness function is changed to:

$$fitness = \max\left(\left|H\left(e^{j\omega i}\right)\right|\right)/H_{target} + \max\left(\left|P_d\left(e^{j\omega i}\right) - P_P\left(e^{j\omega i}\right)\right|\right)/P_{target} \quad (10)$$

where $i = 1\sim L$ is the selected sampling point; $H\left(e^{j\omega i}\right)$ is the amplitude error between the designed filter and the ideal filter at sampling point $i$; $\left|P_d\left(e^{jwi}\right)\right|$ is the phase of the designed filter at sampling point $i$; $\left|P_P\left(e^{jwi}\right)\right|$ is the phase of the ideal filter at sampling point $i$; $H_{target}$ is the weight adjustment parameter of the amplitude; $P_{target}$ is the weight adjustment parameter of the phase.

The design constraints for the amplitude is:

$$\left|H_d\left(e^{j\omega i}\right) - H_P\left(e^{j\omega i}\right)\right| \leq H_{target} \quad (11)$$

In order to satisfy the constraint of Formula (11), the $H\left(e^{j\omega i}\right)$ in the fitness function is defined as follows:

$$\begin{cases} \left|H\left(e^{j\omega i}\right)\right| = \left|H_d\left(e^{j\omega i}\right) - H_P\left(e^{j\omega i}\right)\right|, & if \left|H_d\left(e^{j\omega i}\right) - H_P\left(e^{j\omega i}\right)\right| \leq H_{target} \\ \left|H\left(e^{j\omega i}\right)\right| = \left|H_d\left(e^{j\omega i}\right) - H_P\left(e^{j\omega i}\right)\right| * k, & if \left|H_d\left(e^{j\omega i}\right) - H_P\left(e^{j\omega i}\right)\right| > H_{target} \end{cases} \quad (12)$$

where $k$ is the penalty factor, which is a large integer.

## 3. Results

### 3.1. Filter Design Result of the EA Method

Taking the design of an IIR filter with $m = 6$, $\omega_s = 300$ Hz, and $\omega_p = 400$ Hz as an example, it is needed to design better filter parameters to make the filter indicators $\delta_p$ and $\delta_s$ smaller. Similar to the flow shown in Figure 2, GA, the PSO, PIO, and SMPIO methods are used to design the filter. In this paper, various EA methods have been calculated and compared many times. The parameters are shown in Table 2, and the representative results are shown in Figure 4.

**Table 2.** Evolutionary algorithm methods and the parameters.

| Methods | Parameters |
|---------|-----------|
| GA | particle swarm size: 100, number of generations evolved: 1000, mating probability: 0.5, mutation probability: 0.2 |
| PSO | particle swarm size: 100, number of generations evolved: 1000, inertia weight: $w = 0.8$, acceleration constants: $c1 = 2$, $c2 = 2$ |
| PIO | particle swarm size: 100, number of generations evolved: 1000 $R = 0.001$ |
| SMPIO | particle swarm size: 100, number of generations evolved: 1000 $R = 0.001$, $n_{select} = 50$, $k = 0.0005$ |

The calculation results and the comparison of Figure 4 show that:

- All 4 kinds of EA methods can be used to design IIR filters;
- The PIO method achieves higher accuracy results and faster convergence than the other 3 methods;
- The SMPIO method has the fastest convergence speed and can achieve the highest accuracy.

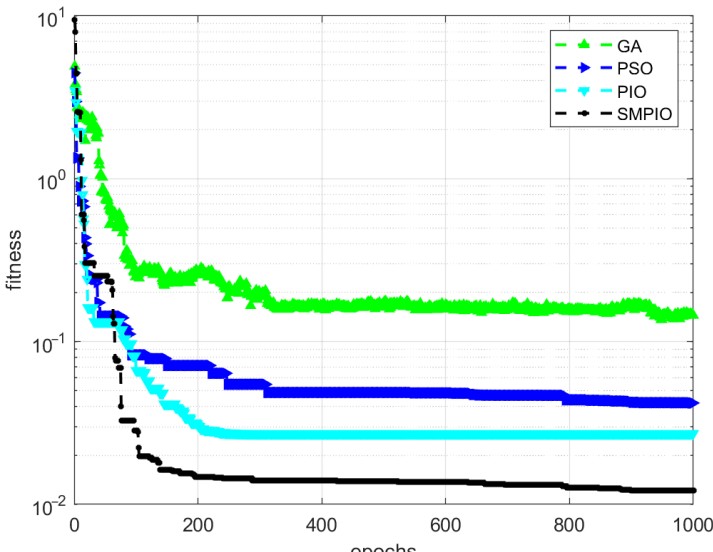

**Figure 4.** Fitness comparison of the GA, PSO, PIO, and SMPIO methods.

Taking the design of an IIR filter with $\omega_s = 300$ Hz, $\omega_p = 400$ Hz, $\delta_p < 0.01$, and $\delta_s < 0.01$ as an example, the orders of the Chebyshev 1 filter, the Chebyshev 2 filter, and the elliptical filter satisfying this condition are 9, 9, and 6, respectively. The elliptical IIR filter with a lower order is selected, considering the calculation ability. On the basis of an IIR filter, the SMPIO method is used to further optimize the elliptical IIR filter. Additionally, taking the design of an IIR filter with $m = 6$, $\omega_s = 300$ Hz, and $\omega_p = 400$ Hz as an example, the amplitude frequency response of the filters are shown in Figure 5. The seven-order elliptical filter can better satisfy the indicators requirements as compared to the seven-order Chebyshev 1 filter and the seven-order Chebyshev 2 filter; the elliptical filter optimized by the SMPIO method obtains a better result. The $\delta_p$ and $\delta_s$ of the optimized filter designed by SMPIO method are reduced so as to improve the performance. On the other hand, a lower-order IIR filter can be obtained under the condition of satisfying $\delta_p < 0.01$ and $\delta_s < 0.01$ by using the SMPIO method.

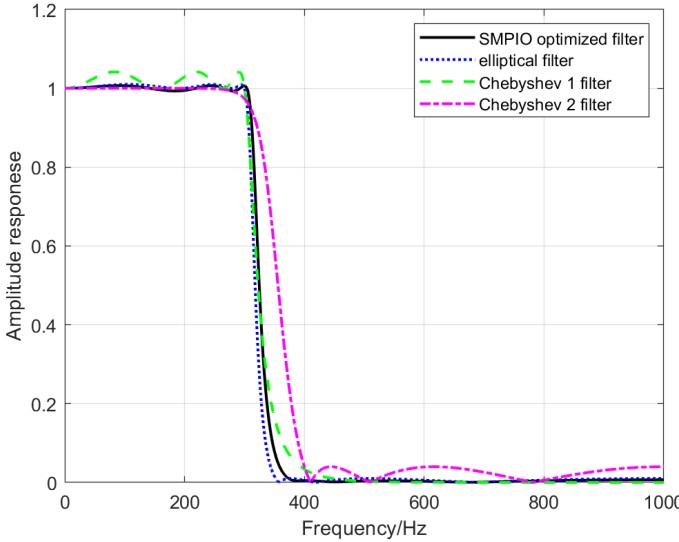

**Figure 5.** Amplitude response comparison of various types of filters (0 Hz~1000 Hz, $m = 6$, amplitude optimized).

The amplitude responses of SMPIO optimized filter (amplitude optimized), elliptical filter, Chebyshev 1 filter, and Chebyshev 2 filter with $\omega_s = 300$ Hz, $\omega_p = 400$ Hz, and $m = 6$ are show in Figures 5–7. Figure 5 shows the amplitude responses of the four

filters in the 0–1000 Hz range. To display the details of Figure 5 more clearly, we add Figures 6 and 7, where Figure 6 more clearly shows the amplitude responses of the four filters in the 0–300 Hz range, and Figure 7 more clearly shows amplitude responses in the range 400–1000 Hz. It can be seen from Figures 5–7 that the absolute value of the amplitude response error of the SMPIO optimized filter is the smallest; that is, $\delta_p$ and $\delta_s$ is the smallest, and the result is the best. The IIR filter parameters are show in Table 3.

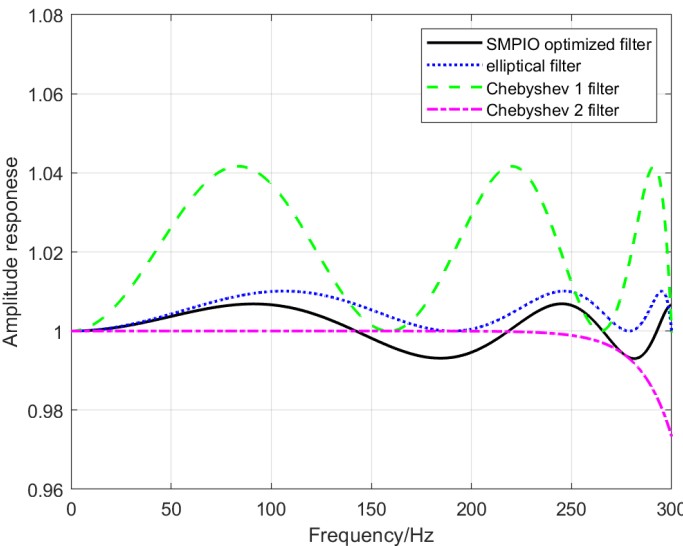

**Figure 6.** Amplitude response comparison of various types of filters (0 Hz~300 Hz, $m = 6$, amplitude optimized).

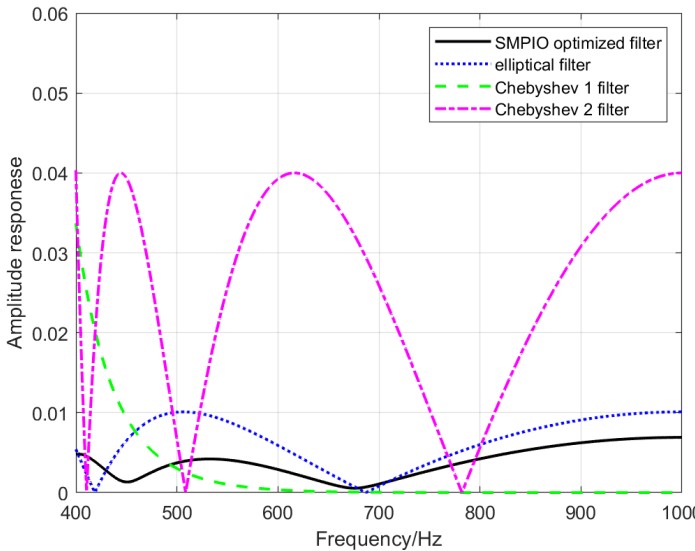

**Figure 7.** Amplitude response comparison of various types of filters (400 Hz~1000 Hz, $m = 6$, amplitude optimized).

LINS can be applied to highly maneuverable carriers such as tactical missiles for real-time attitude control. In this case, the INS filter not only requires the amplitude to meet certain indicators, but also requires a small phase. The LINS filter should meet the following comprehensive requirements: jitter filtering performance (including passband frequency band, and allowable error) for laser gyro accuracy; less filter order to adapt to real-time fast calculation requirements; less phase delay for high dynamic control.

Taking the design of an IIR filter as an example, the engineering application requirements are:

- condition A, $\omega_s = 300$ Hz, $\omega_p = 400$ Hz, $\delta_p < 0.02$, and $\delta_s < 0.02$;

- condition B, the order does not exceed 6;
- condition C, the phase of 0 Hz~100 Hz does not exceed −40 degree.

**Table 3.** Various types of IIR filter parameters ($\omega_s$ = 300 Hz, $\omega_p$ = 400 Hz, *m* = 6, amplitude optimized).

| Filter Types | IIR Filter Parameters |
|---|---|
| SMPIO optimized filter | *a* = [1.0000000000, −3.1469411000, 5.1902995000, −5.1231126000, 3.1948383000, −1.1687601000, 0.1983537300] <br> *b* = [0.0226377066, −0.0004920049, 0.0484860398, 0.0059140339, 0.0457274571, 0.0012101968, 0.0211943007] |
| elliptical filter | *a* = [1.0000000000, −3.1409901000, 5.1933596000, −5.1225648000 3.1976146000, −1.1680883000, 0.1992300800] <br> *b* = [0.0300495577, −0.0078353814, 0.0576256820, −0.0011186365, 0.0576256820, −0.0078353814, 0.0300495577] |
| Chebyshev 1 filter | *a* = [1.0000000000, −3.6736367000, 6.5299764000, −6.8973626000, 4.5129637000, −1.7260817000, 0.3022655200] <br> *b* = [0.0007519472, 0.0045116830, 0.0112792081, 0.0150389434, 0.0112792081, 0.0045116830, 0.0007519472] |
| Chebyshev 2 filter | *a* = [1.0000000000, −1.2474697000, 1.5353083000, −0.7027603300, 0.3605929200, −0.0372448440, 0.0134976460] <br> *b* = [0.0874795614, 0.0917752434, 0.1919701708, 0.1794740408, 0.1919701708, 0.0917752434, 0.0874795614] |

Design the IIR filter with only condition A as an index: the orders of the Chebyshev 1 filter, the Chebyshev 2 filter, and the elliptical filter satisfying this condition are 8, 8, and 5, respectively; the max phases of 0~100 Hz of the Chebyshev 1 filter, the Chebyshev 2 filter, and the elliptical filter satisfying this condition are −40.3 degree, −40.3 degree, and −46.2 degree, respectively, which is shown in Figure 8. The Chebyshev 1 filter, the Chebyshev 2 filter, and the elliptical filter cannot satisfy condition A, condition B, and condition C at the same time. The Chebyshev 1 filter, and the Chebyshev 2 filter can satisfy condition A and condition C basically, but cannot satisfy condition B. The elliptical filter can satisfy condition A and condition B basically, but cannot satisfy condition C.

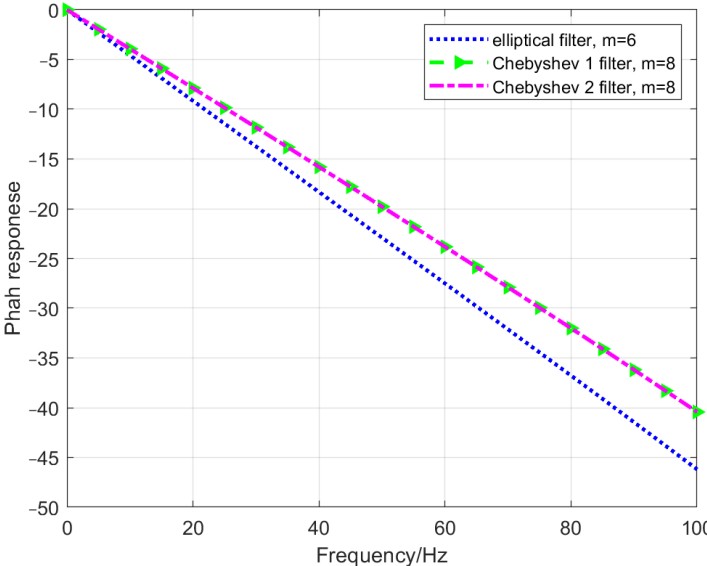

**Figure 8.** Phase response comparison of various types of filters (0 Hz~100 Hz).

Taking the design of an IIR filter with *m* = 6, the SMPIO method is used to get a optimized filter according to the flow chart of Figure 3 with the fitness function of Formula (10), and the design constraints of Formulas (11)–(12). The amplitude responses of SMPIO optimized filter (amplitude and phase optimized), elliptical filter, Chebyshev 1 filter,

and Chebyshev 2 filter with $\omega_s = 300$ Hz, $\omega_p = 400$ Hz, and $m = 6$ are show Figures 9–11, and the phase responses with 0 Hz~100 Hz are show in Figure 12. Figure 9 shows the amplitude responses of the four filters in the 0–1000 Hz range. To display the details of Figure 9 more clearly, we add Figures 10 and 11, where Figure 10 more clearly shows the amplitude responses of the four filters in the 0–300 Hz range, and Figure 11 more clearly shows amplitude responses in the range 400–1000 Hz. The SMPIO optimized filter can satisfy condition A, condition B, and condition C at the same time. The Chebyshev 1 filter cannot satisfy condition A and condition B. The Chebyshev 2 has a smaller phase, but cannot satisfy condition A. The elliptical filter can satisfy condition A and condition B, but cannot satisfy condition C. For engineering applications that need to meet the requirements of conditions A, B, and C, only the SMPIO optimized filter can meet the requirements of the comprehensive design of the amplitude and phase simultaneously; the SMPIO optimized filter has the best performance. The IIR filter parameters are show in Table 4.

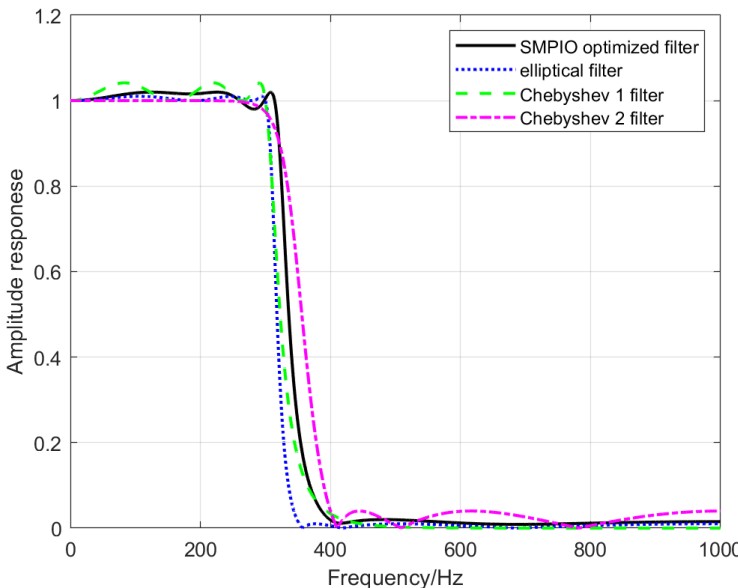

**Figure 9.** Amplitude response comparison of various types of filters (0 Hz~1000 Hz, *m* = 6, amplitude and phase optimized).

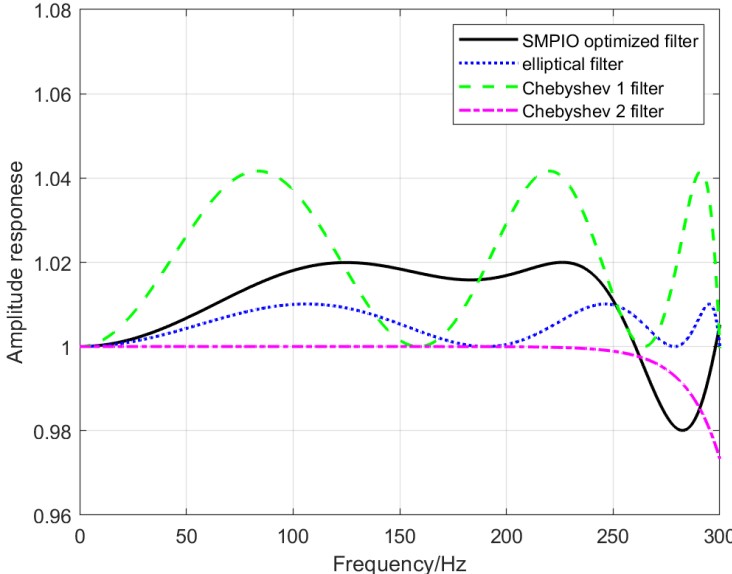

**Figure 10.** Amplitude response comparison of various types of filters (0 Hz~300 Hz, *m* = 6, amplitude and phase optimized).

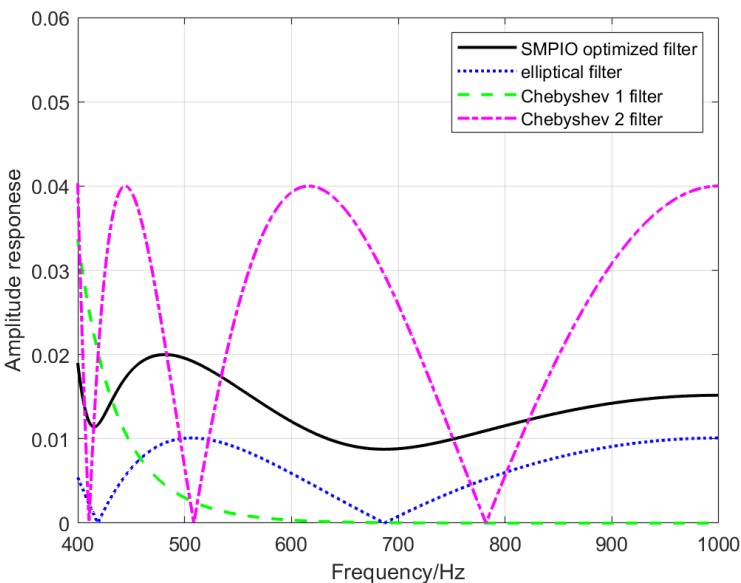

**Figure 11.** Amplitude response comparison of various types of filters (400 Hz~1000 Hz, *m* = 6, amplitude and phase optimized).

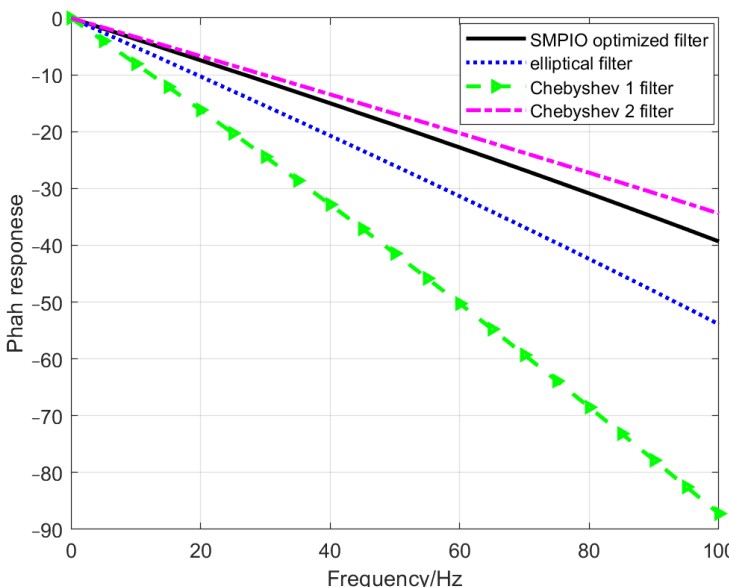

**Figure 12.** Phase response comparison of various types of filters (0 Hz~100 Hz, *m* = 6, amplitude and phase optimized).

**Table 4.** Various types of IIR filter parameters ($\omega_s$ = 300 Hz, $\omega_p$ = 400 Hz, *m* = 6, amplitude and phase optimized).

| Filter Types | IIR Filter Parameters |
| --- | --- |
| SMPIO optimized filter | $a$ = [1.0000000000, −3.1464571000, 5.1854756000, −5.1254502000, 3.2072909000, −1.1781249000, 0.2027716000] <br> $b$ = [0.0636898053, −0.0376032393, 0.0900215835, −0.0220216831, 0.0467710785, −0.0120861956, 0.0167345507] |

### 3.2. Effect of LINS Filter

The IIR-type low-pass filter is selected for filtering by a laser inertial navigation system after considering the instrument accuracy, calculation capacity, vibration frequency, and system dynamic characteristics. The SMPIO method described above can be used for LINS

filter optimization. The data in Figure 2 are filtered by the optimized IIR filter, which was designed using the SMPIO method. The laser gyroscope data obtained after filtering and considering the frequency characteristics are shown in Figure 13.

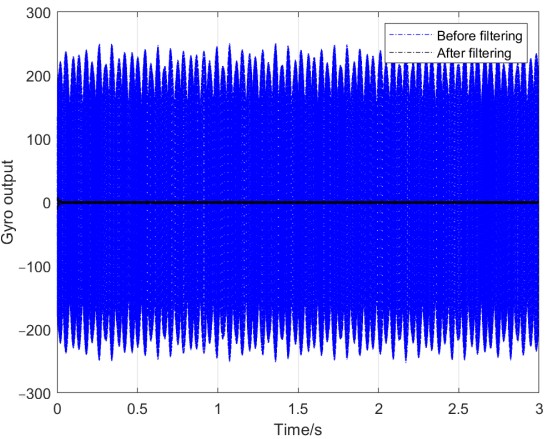

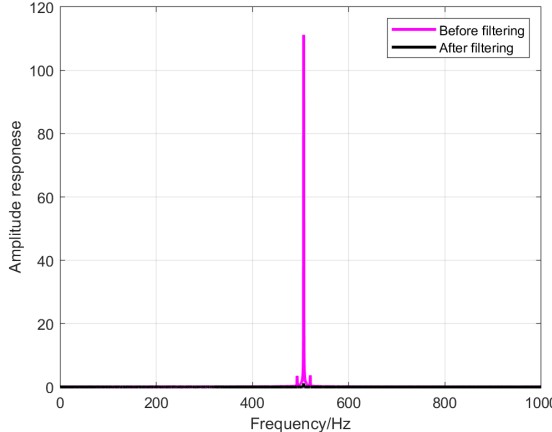

**Figure 13.** The original information and spectrum analysis results of the laser gyro before and after filtering ($f_s = 2 \, \text{k Hz}$, $f_D = 515 \, \text{Hz}$).

## 4. Discussion

The PIO method proposed in this paper for the design of filters for laser inertial navigation system can flexibly set the filter indicators. Here, an optimal filter design for laser inertial navigation system is achieved under the comprehensive consideration of instrument accuracy, calculation capacity, vibration frequency, system dynamic characteristics, and other indicators, and it can be used for laser gyroscope filtering, which has great engineering significance. Compared with the results of the GA, PSO, PIO, and other evolutionary algorithms, the PIO algorithm has higher accuracy and convergence speed. We propose an improved pigeon-inspired optimization method based on natural selection and Gaussian mutation. The calculation results show that the SMPIO method has higher accuracy and stability. SMPIO method can flexibly design filters according to the comprehensive requirements of laser gyro performance and navigation control indicators, which cannot be achieved by traditional filter design methods.

The scope discussed in this paper is limited to the problem of the dither wave filter of low precision laser gyro, so the design indicators of the amplitude are low standard, and the IIR filter has a lower order ($\delta_p < 0.02$, $\delta_s < 0.02$, and $m = 6$). For high-precision laser gyro, the requirements for amplitude design indicators will increase, and the order of the IIR filter will increase accordingly (such as $\delta_p < 0.001$, $\delta_s < 0.001$, and $m = 10$). With the increase of the filter order, the parameters to be optimized of SMPIO increase, and the

optimization of the filter becomes more difficult. Considering the application requirements of real-time fast calculation, this paper focuses on IIR filters. In the future, we will continue to verify the optimization effect of the SMPIO method for FIR filters.

**Author Contributions:** Conceptualization, Methodology, Investigation, Software, Writing—original draft: Z.L.; Writing—review and editing, Funding acquisition, Project administration: L.Z.; Data curation, Resources: K.W.; All authors have read and agreed to the published version of the manuscript.

**Funding:** This research received no external funding.

**Data Availability Statement:** The data presented in this study are available on request from the corresponding author.

**Conflicts of Interest:** The authors declare no conflicst of interest.

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
