# Peer review of "Filter Design for Laser Inertial Navigation System Based on Improved Pigeon-Inspired Optimization"

_aerospace, doi:10.3390/aerospace10010063_

Round 1

Reviewer 1 Report

In this paper, the authors proposed the PIO (pigeon-inspired optimization) method for the design of filters for laser inertial navigation system.

The authors in the introduction did not outline the research gap. Only a comprehensive review of the literature on the topic at hand is presented.

The paper is interesting, and I have no fundamental doubts about its content. However, I found some errors and inaccuracies and the authors should correct them.

 Weak

In my opinion, a weakness of the work is the insufficient description of the state of the issue with only 28 references to the literature, in addition, only 21 percent of the bibliographic items are publications no older than 3 years. Supplementation in this regard would certainly increase the value of the paper.

 Noticed errors/remarks

          The Introduction chapter lacked a summary of the issue status analysis. Nor was the research gap clearly presented as a direct result of the analysis of the state of the issue.

          Line 44 contain "wholesale" numbers of literature references without any, even a brief, characterization of each. This is not correct practice for more than 2 to 3 references.

          I also did not find a definition of the term dither in the article. It is an unwanted signal of noise nature? If so, with what parameters?

          Line 304. What is Figure 9~Figure 11 mean???

          Line 352: What does the term “low-oder” mean?

          In my opinion, figures 7 and 8 are described in too small a font in relation to the whole text. This should be corrected. This will certainly improve the readability of the entire work.

          The literature list deviates slightly from the journal template. It should be carefully adjusted according to the requirements.

 Small errors

I hope a typographic type only. They do not diminish the value of work, but they must be corrected.

·        Throughout the paper, standardize the notation of variables appearing in equations and text by italicizing them.

·        Insert a space between the value and the unit. Applies to the entire work.

·        Line 10 Is: “System(LINS)”, should be: “System (LINS)”. The spaces are missing. Please correct throughout the paper.

·        Lina 27 Is: “mutation(SMPIO) ,”, should be:” mutation (SMPIO),”. The space character should be before the parenthesis, not after it.

·        Line 38 Is: “gyroscopes, and”, should be: “gyroscopes and“. I'ts better to have no comma between these phrases.

·        Line 38 Is: “acceleration[1, 2]”, should be: “acceleration [1, 2]“. The spaces are missing. Please correct throughout the paper.

·        Line 120. Is: whith, should be: with

·        Line 244. Is: ofFigure 4, should be: of Figure 4

Author Response

Thanks for your thoughtful and helpful comments, we sincerely appreciated your great help.

Reviewer 2 Report

1. I like your idea of putting all abreviations together in the table in the very beginning. However, some of them are still introduced once again in text (and some are not). Also, some abbreviations are arguable. For example, why evolutionary opitmization is abbreviated as EA? The same question for MPIO and DVPIO.

2. I think it would be better to finish the introduction with a couple of sentences introducing the main scope of your research, em[phasizing on the problem that is about to be solved. 

3. I do not really understand the following part of the sentence: "LINS must carry out .... inertial navigation, integrated navigation, and visual navigation, ...". What do you call  inertial navigation, integrated navigation, and visual navigation in this context?

4. In formula (5) (which is, by the way, probably numbered in the wrong line) you generally say that Velocity = Velocity + Position. In Formula (6) you generally say that Position = Position + Velocity. I assume that in this case these variables are related to some mathematical abstracts instead of the physical meaning of position and velocity, but, still, please, check if everything is correct here. 

5. In line 163: ...particles with the largest fitness are eliminated and replaced by the ... particles with the best fitness. If you talk about the largest and the smallest fitness (according to line 200. for example), than why do you adress to "the best" fitness here? If the largest is the best than what substitution are we making?

6. I think it is worth to explain in more details why do you state that MPIO optimized filter is better than elliptical after figures 5-7 and 9-11.

Author Response

(The authors gave the same response as above.)
